# Selenium intake in relation to all-cause and cardiovascular mortality in individuals with nonalcoholic fatty liver disease: A nationwide study in nutrition

Xin Dong [iD][‡], Yunchao Deng, Gang Chen[iD]*

Department of Cardiovascular Medicine, The Central Hospital of Wuhan, Tongji Medical College, Huazhong University of Science and Technology, Wuhan, China

‡ XD share first authorship on this work.
* tomgangchen@sohu.com

## Abstract

### Aims

Limited evidence exists regarding the association of selenium with risk of death in individuals with nonalcoholic fatty liver disease (NAFLD). This study was designed to investigate the relationship between dietary selenium intake with mortality in a nationally representative sample of United States adults with NAFLD.

### Methods

Dietary selenium intake was assessed in 2274 NAFLD adults younger than 60 years of age from the National Health and Nutrition Examination Survey (NHANES) III through a 24-hour dietary recall. NAFLD was diagnosed by liver ultrasound after excluding liver disease due to other causes. Cox proportional hazards models were utilized to assess the effect of dietary selenium intake on all-cause and cardiovascular mortality among individuals with NAFLD.

### Results

At a median follow-up of 27.4 years, 577 deaths occurred in individuals with NAFLD, including 152 cardiovascular deaths. The U-shaped associations were discovered between selenium intake with all-cause ($P_{nolinear} = 0.008$) and cardiovascular mortality ($P_{nolinear} < 0.001$) in adults with NAFLD after multivariate adjustment, with the lowest risk around selenium intake of 121.7 or 125.9 µg/day, respectively. Selenium intake in the range of 104.1– 142.4 µg/day was associated with a reduced risk of all-cause mortality and, otherwise, an increased risk. Selenium intake in the range of 104.1–150.6 µg/day was associated with a reduced risk of cardiovascular death and, otherwise, an increased risk.

**Data Availability Statement:** All relevant third-party data utilized during the course of this study are publicly available from the CDC NCHS NHANES repository (https://www.cdc.gov/nchs/nhanes).

**Funding:** The author(s) received no specific funding for this work.

**Competing interests:** The authors have declared that no competing interests exist.

## Conclusions

Both high and low selenium intake increased the risk of all-cause and cardiovascular death in adults younger than 60 years of age with NAFLD, which may help guide dietary adjustments and improve outcomes in adults with NAFLD.

## Introduction

In recent years, NAFLD has been recognized as one of the most prevalent liver diseases worldwide, affecting approximately 30% of the world's population [1]. NAFLD is defined as the excessive accumulation of lipids in liver cells in the absence of alcohol abuse or hepatitis virus infection [2]. NAFLD can not only progress to non-alcoholic steatohepatitis (NASH) and liver fibrosis, but also cause complications such as obesity, dyslipidemia, insulin resistance and/or diabetes, and hypertension, and even hepatocellular carcinoma [3]. Given the common risk factors between NAFLD and cardiovascular disease, cardiovascular death is one of the leading causes of death among individuals with NAFLD [4]. Since there is currently no recognized treatment for NAFLD, lifestyle intervention and dietary modification are important ways to prevent the progression of NAFLD.

The pathogenesis of NAFLD is mainly related to hepatic lipid deposition caused by insulin resistance and mitochondrial dysfunction caused by oxidative stress [5]. Selenium is an essential element that plays an important role in redox homeostasis, thyroid hormone metabolism, defense against oxidative stress and inflammation [6]. Selenium is mainly derived from food, rich in organ meat and seafood, followed by grains, cereals and dairy products [7]. Epidemiological studies have shown that both high and low selenium intake can harm health [6]. However, due to limited epidemiological evidence on dose-response relationships between selenium and specific health outcomes, there is still a lack of consensus on reference levels for selenium intake [6].

There is growing evidence from recent observational studies and randomized clinical trials that high selenium exposure may have adverse effects on NAFLD, cardiometabolic health, particularly hypertension, dyslipidemia, and diabetes [8–11]. Reja et al. found that serum selenium levels were inversely associated with the risk of advanced liver fibrosis and all-cause mortality in patients with NAFLD [12]. This study mainly reflected the effect of blood selenium homeostasis on liver fibrosis in patients with NAFLD, rather than dietary selenium intake. However, the associations and dose-response relationships between dietary selenium intake and all-cause and cardiovascular mortality in individuals with NAFLD are unclear. The primary objective of our study was to investigate the association between dietary selenium intake with all-cause and cardiovascular mortality in adults under than 60 years of age with NAFLD using the NHANES III collected from 1988–1994. The secondary goal is to determine the optimal range for dietary selenium intake, which may help guide dietary adjustments in patients with NAFLD. The third objective is to investigate the effects of dietary selenium intake on glucose metabolism and lipid metabolism in adults under 60 years of age with NAFLD.

## Methods

### Study population

The National Health and Nutrition Examination Survey (NHANES)-III (1988–1994) is a survey research program in the United States to assess the health status of the civilian population. All data and materials have been made publicly available at the National Center for Health

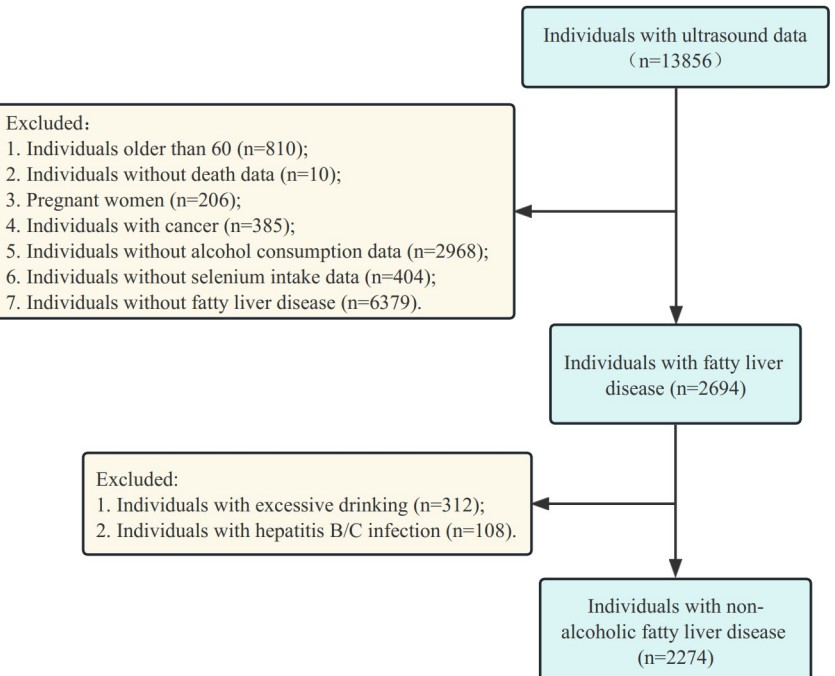

**Fig 1. Flow chart of the study participants.**

Statistics website (https://www.cdc.gov/nchs/nhanes/index.htm). The protocols for NHANES (https://www.cdc.gov/nchs/nhanes/index.htm) were approved by the National Center for Health Statistics of the Centers for Disease Control (CDC) and Prevention Institutional Review Board, and Institutional Review Board (IRB) approval and documented consent was obtained from all participants.

Participants were 12,642 adults over the age of 20 and under the age of 60 who had liver ultrasound and selenium intake data. The study included 2,274 adults younger than 60 years of age with NAFLD. After excluding participants with missing variables and ineligible for diagnosis, 2274 participants with NAFLD were eventually included in the analysis. Flow chart of the study participants is shown in Fig 1.

## Selenium measurements

Dietary selenium intake was measured by 24-hour dietary recall interviews. During the interview, participants provided details (including descriptions, quantities, and nutritional content) of the foods and beverages consumed during the 24 hours prior to the interview (midnight to midnight). For each participant, the amount of nutrients consumed from each food or beverage, including selenium intake, was calculated using United States Department of Agriculture Dietary Research Food and Nutrition Database [13]. Nutritional estimates do not include nutrients obtained from dietary supplements or medications. Protocols and data collection methods are documented elsewhere [14].

## NAFLD assessment

Transient elastography examinations were performed for all participants aged 20 years and older in NHANES III. Patients with hepatic steatosis (including mild, moderate, and severe

cases) are identified as having fatty liver disease (FLD). NAFLD was diagnosed in patients with FLD without hepatitis B/C infection or heavy alcohol consumption (≥2 drinks/day for men or ≥1 drink/day for women) [15].

## Ascertainment of outcomes

National Center for Health Statistics mortality was determined using a probabilistic record match between NHANES participants and National Death Index (NDI) death certificate data. Through December 31, 2019, the NHANES-linked NDI public access data was used to identify the mortality status and cause of death. Data on the leading cause of death were used for case definition according to the codes of the International Classification of Diseases 10th Revision (ICD-10). Cardiovascular mortality included deaths from heart diseases (ICD-10 codes I00-I09, I11, I13, I20-I51) or cerebrovascular diseases (ICD-10 codes I60-I69).

## Covariates

Age (in years), sex (binary: male and female), race or ethnicity (categorical: non-Hispanic white, non-Hispanic black, Mexican American, other Hispanic, and other race or multiracial); educational level (categorical: less than high school, high school graduate or equivalent, college graduate or higher) were self-reported by the participants. Body Mass Index (BMI), the ratio of weight-to-height$^2$ measured in kg/m$^2$, was available as a continuous measure. Poverty income ratio (PIR) was calculated as the ratio of monthly family income to poverty levels and categorized into 3 groups: < 1.3 (low income), 1.3–3.5 (middle income), and > 3.5 (high income). We categorized smoking into three categories: never smoker (smoked <100 cigarettes during their lifetime), former smoker (smoked ≥100 cigarettes during lifetime but did not smoke at the time of interview), and current smoker (smoked ≥100 cigarettes during lifetime and reported smoking at the time of interview). Physical activity (categorical: inactive, insufficiently active, active) was assessed by Global Physical Activity Questionnaire, which includes the frequency and duration of physical activity performed during daily activities, leisure time activities, and sedentary activities. Dietary supplement use (dichotomized, yes or no) was based on participants' self-reported use of any vitamins, minerals, or dietary supplements in the past month.

Participants were also classified in terms of their health outcomes. Diabetes was defined as the previous diagnosis of diabetes, current use of diabetic pills or insulin, a hemoglobin A1c level of ≥6.5%, a fasting plasma glucose level of ≥126mg/dL, or 2-h plasma glucose level of ≥200mg/dL [16]. Hypertension was defined as self-reported hypertension diagnosed by doctors, systolic blood pressure ≥140 mmHg or diastolic blood pressure ≥90 mmHg, or specific drug treatment [17]. Dyslipidemia was diagnosed if plasma Triglycerides (TGs) ≥150 mg/dL, plasma total cholesterol ≥200 mg/dL, plasma Low Density Lipoprotein (LDL) cholesterol ≥130 mg/dL, specific drug treatment, self-reported high blood cholesterol diagnosed by doctors, or high density lipoprotein (HDL) cholesterol <40 mg/dL for males or <50 mg/dL for females [18]. The Fibrosis-4 index was a noninvasive method to assess the degree of liver fibrosis, calculated by age, aspartate and alanine aminotransferase concentrations, and platelet counts [19]. Cardiovascular disease was a self-reported diagnosis of congestive heart failure, coronary heart disease, angina pectoris, heart attack, and stroke. Blood samples were collected and sent to central laboratories for the determination of blood lipids, plasma glucose, C-reactive protein (CRP), uric acid, alanine aminotransferase (ALT), aspartate aminotransferase (AST), and hemoglobin A1c (HbA1c).

## Statistical analysis

All analyses were performed accounting for the complex survey design using the appropriate subsample weights, strata, and primary sampling units per NHANES recommendation.

An initial descriptive analysis was conducted to examine considerable differences in baseline demographic and life factors by Chi-square test for categorical variables and linear regression model for continuous variables. Population characteristics are expressed as the mean ± Standard Deviation (SD) or numbers of observations (percent). The generalized weighted linear model was applied to examine the associations of dietary selenium intake with cardiometabolic biomarkers and markers of liver function impairment at baseline, including glucose, HbA1c, homeostasis model assessment of insulin resistance (HOMA-IR), triglycerides, HDL-C, LDL-C, CRP, uric acid, ALT and AST.

We firstly performed test for proportional hazards assumption before Cox regression models were fitted. All models satisfied the proportional hazards assumption since the graph of the survival function versus the survival time resulted in graphs with parallel curves. Cox proportional hazards models were adopted over time in the study as the underlying time-axis to calculate hazard ratios (HRs) and the corresponding 95% confidence intervals (CIs) to ascertain the association between dietary selenium intake with all-cause and cardiovascular mortality in adults with NAFLD. Dietary selenium intake, as the independent variable, was modeled categorically (quartiles) and continuously (log-transformed value), respectively. In Model 1, age (years), sex (male or female), and self-reported race (non-Hispanic White, non-Hispanic Black, Mexican American, or others) were adjusted; Model 2 further adjusted (from Model 1) for education (less than high school, high school or equivalent, or college or above), family income-poverty ratio ($\leq$1.30, 1.31–3.50, or >3.50), Healthy Eating Index (HEI) (continuous), smoking status (never, former, or current smoker), physical activity (inactive, insufficiently active, or active), BMI (kg/m$^2$; <25.0, 25.0–29.9, or $\geq$30.0), diabetes (yes or no), hypertension (yes or no), dyslipidemia (yes or no), Fibrosis-4 index (<1.30, 1.30–2.66, $\geq$2.67). Furthermore, the non-linear association of interest (on a continuous scale) was examined by the restricted cubic spline regression models with three knots (10th, 50th, and 90th) after adequate adjustment for confounders.

Stratified analyses were also performed to explore underlying effect modification, including gender, race/ethnicity (non-Hispanic white, others), BMI (<25 kg/m$^2$, $\geq$25 kg/m$^2$), smoking status (yes, no), physical activity (yes, no), dyslipidemia (yes, no), diabetes (yes, no) and hypertension (yes, no). The potential interactions between dietary selenium intake and these stratifying variables were detected by adding their cross-product terms into the model accordingly.

In addition, to check the robustness of the association, a series of sensitivity analyses were performed as follows: (1) considering that heart failure or stroke often leads to worse survival, we excluded subjects with a history of heart failure or stroke; (2) we also removed those whose deaths occurred during the first five years of follow-up to avoid potential reverse causality as much as possible.

All analyses were carried out in R version 4.0.5. *P*-values were corrected for multiple testing using Benjamini-Hochberg method for false discovery rate (FDR) control [20], with two-tailed $P < 0.05$ being regarded as statistically significant.

## Results

### Description of study participants

There were 2274 NAFLD patients older than 20 years and younger than 60 years included in current study, of which approximately 50.5% were male. The baseline characteristics of

participants by quartile of dietary selenium intake are presented in Table 1. The mean age of all included patients was 39.60 ± 10.94 years old. Subjects with higher dietary selenium intake were female, more likely to be non-Hispanic, had higher household income and HEI scores. As shown in the S1 Table, participants with higher dietary selenium intake were positively associated with higher levels of LDL-C and HOMA-IR (all $P_{trend} < 0.05$).

## Relationship between dietary selenium intake and mortality

During a median period of 27.4 years, a total of 577 deaths occurred, including 152 deaths from cardiovascular diseases. Kaplan–Meier curves showed that participants in Quintile 1 of dietary selenium intake had lower cumulative survival probability (Fig 2A and 2B). Table 2 lists the relationships between dietary selenium intake and risk of all-cause and cardiovascular mortality among patients with NAFLD in Cox proportional regression models adjusted for covariates. With adjustments for covariates, dietary selenium intake showed U-shaped associations with the risk of all-cause and cardiovascular mortality. Compared with participants in selenium intake Quartile 1, significantly lower risks of all-cause mortality were found in Quartile 2 (HR, 0.61; 95% CI: 0.43–0.87), Quartile 3 (HR, 0.63; 95% CI: 0.39–1.01), and Quartile 4 (HR, 0.67; 95% CI: 0.48–0.95), and significantly lower risks of cardiovascular mortality were found in Quartile 2 (HR, 0.19; 95% CI: 0.08–0.45), Quartile 3 (HR, 0.29; 95% CI: 0.12–0.68), and Quartile 4 (HR, 0.38; 95% CI: 0.17–0.85). We also found that for every one unit increase in natural logarithmic conversion of dietary selenium intake, the risk of cardiovascular mortality was reduced by 72% (HR, 0.28; 95% CI: 0.09–0.91) (Table 2).

## Non-linear dose-response relationship between dietary selenium intake and mortality

As shown in Fig 2C and 2D, we used Cox proportional hazard regression models with restricted cubic spline regression to find U-shaped associations between dietary selenium intake and all-cause mortality and cardiovascular mortality in NAFLD patients (all $P_{nonlinear} < 0.05$). Then, two inflection points are calculated by recursive algorithm, which are 121.7 μg/d and 125.9 μg/d, respectively. In addition, a two-stage Cox proportional risk model was used on both sides of the inflection point to verify the nonlinear U-shaped relationship between dietary selenium intake and all-cause mortality and cardiovascular mortality ($P$ for log-likelihood ratio < 0.05) (Table 3). When dietary selenium intake was less than 125.9 μg/day, cardiovascular mortality was decreased by 2% for each Standard Deviation (SD) increase in dietary selenium intake (HR 0.98; 95% CI 0.98–0.99; $P = 0.001$). When dietary selenium intake was more than 125.9 μg/day, cardiovascular mortality was increased by 1% for each SD increase in dietary selenium intake (HR 1.01; 95% CI 1.00–1.02; $P = 0.007$) (Table 3). Selenium intake in the range of 104.1–142.4 μg/day was associated with a reduced risk of all-cause mortality and, otherwise, an increased risk. Selenium intake in the range of 104.1–150.6 μg/day was associated with a reduced risk of cardiovascular death and, otherwise, an increased risk.

## Stratified analyses

Stratified analyses were further conducted to evaluate the association of selenium intake with all-cause (<104.1 vs. 104.1–142.4 vs. >142.4 μg/day) and cardiovascular mortality (<104.1 vs. 104.1–150.6 vs. >150.6 μg/day) in various subgroups. The results of the stratified analysis are shown in Figs 3 and 4. Such U-shaped associations remained when analyses were stratified by gender (male or female), race/ethnicity (non-Hispanic white or other), BMI (<25 kg/m$^2$, ≥25 kg/m$^2$), smoking status (yes, no), physical activity (yes, no), dyslipidemia (yes, no), diabetes (yes, no) and hypertension (yes, no), even though in some strata, the associations became

**Table 1. Baseline characteristics of participants with NAFLD according to selenium intake (quartiles) in NHANES III.**

| | | Selenium intake (μg/day) | | | | P-value |
|---|---|---|---|---|---|---|
| | Overall | <78.0 | 78.0–111.5 | 111.6–150.9 | >150.9 | |
| **Participants, n (%)** | 2274 | 656 (25) | 599 (25) | 495 (25) | 524(25) | |
| **Age (mean (SD))** | 39.60 (10.94) | 40.40 (10.85) | 39.70 (11.40) | 39.99 (10.76) | 38.12 (10.75) | 0.751 |
| **Sex, n (%)** | | | | | | <0.001 |
| Male | 1028 (50.5) | 149 (26.3) | 230 (39.1) | 265 (60.7) | 384 (75.8) | |
| Female | 1246 (49.5) | 507 (73.7) | 369 (60.9) | 230 (39.3) | 140 (24.4) | |
| **Race, n (%)** | | | | | | 0.022 |
| Non-Hispanic White | 714 (71.9) | 172 (64.0) | 194 (75.1) | 186 (76.9) | 162 (71.6) | |
| Non-Hispanic Black | 583 (10.0) | 183 (12.1) | 148 (10.3) | 113 (8.1) | 139 (9.5) | |
| Mexican American | 881 (7.9) | 267 (9.2) | 243 (8.7) | 170 (6.3) | 201 (7.5) | |
| Other | 96 (10.2) | 34 (14.7) | 14 (5.9) | 26 (8.7) | 22 (11.4) | |
| **Family income-poverty ratio, n (%)** | | | | | | 0.018 |
| ≤1.3 | 720 (31.7) | 248 (37.8) | 199 (33.2) | 139 (28.1) | 134 (25.6) | |
| 1.31–3.5 | 1090 (47.9) | 278 (42.4) | 286 (47.7) | 265 (53.5) | 261 (49.8) | |
| >3.5 | 464 (20.4) | 130 (19.8) | 114 (19.0) | 91 (18.4) | 129 (24.6) | |
| **Education, n (%)** | | | | | | 0.073 |
| Less than high school | 891 (24.6) | 301 (30.8) | 233 (22.7) | 176 (20.1) | 181 (24.6) | |
| High school or equivalent | 1117 (56.1) | 292 (54.8) | 304 (59.4) | 251 (56.8) | 270 (53.2) | |
| College or above | 258 (19.3) | 61 (14.4) | 60 (17.9) | 66 (23.1) | 71 (22.1) | |
| **Smoke, n (%)** | | | | | | 0.097 |
| Never smoker | 1205 (48.1) | 392 (57.5) | 331 (47.3) | 247 (42.7) | 235 (44.9) | |
| Ever smoker | 509 (25.5) | 120 (19.7) | 134 (26.2) | 127 (28.9) | 128 (27.3) | |
| Current smoker | 559 (26.4) | 144 (22.8) | 134 (26.6) | 121 (28.4) | 160 (27.8) | |
| **Physical activity, n (%)** | | | | | | 0.259 |
| Inactive | 701 (23.6) | 245 (27.6) | 196 (26.2) | 135 (19.9) | 125 (20.5) | |
| Insufficiently active | 1037 (52.6) | 257 (48.8) | 273 (50.2) | 258 (59.1) | 249 (52.5) | |
| Active | 433 (23.8) | 121 (23.5) | 105 (23.5) | 87 (21.1) | 120 (27.0) | |
| **Body mass index, n (%)** | | | | | | 0.43 |
| <25 | 603 (29.0) | 161 (31.0) | 150 (27.8) | 150 (30.3) | 142 (26.9) | |
| 25–29.9 | 690 (31.4) | 208 (30.0) | 179 (27.9) | 140 (30.8) | 163 (36.9) | |
| ≥30 | 981 (39.6) | 287 (39.0) | 270 (44.3) | 205 (38.9) | 219 (36.1) | |
| **Hypertension, n (%)** | | | | | | 0.915 |
| Yes | 752 (32.6) | 220 (29.8) | 193 (34.6) | 161 (33.3) | 178 (32.6) | |
| No | 1522 (67.4) | 436 (70.2) | 406 (65.4) | 334 (66.7) | 346 (67.4) | |
| **Diabetes, n (%)** | | | | | | 0.243 |
| Yes | 323 (9.3) | 94 (9.8) | 98 (10.8) | 67 (10.1) | 64 (6.5) | |
| No | 1951 (90.7) | 562 (90.2) | 501 (89.2) | 428 (89.9) | 460 (93.5) | |
| **Dyslipidemia, n (%)** | | | | | | 0.623 |
| Yes | 1815 (78.8) | 530 (78.9) | 484 (82.4) | 391 (78.0) | 410 (75.8) | |
| No | 459 (21.2) | 126 (21.1) | 115 (17.6) | 104 (22.0) | 114 (24.2) | |
| **Cardiovascular disease, n (%)** | | | | | | 0.879 |
| Yes | 780 (33.8) | 230 (31.3) | 200 (36.2) | 166 (34.1) | 184 (33.6) | |
| No | 1471 (66.2) | 419 (68.7) | 391 (63.8) | 326 (65.9) | 335 (66.4) | |
| **Stroke, n (%)** | | | | | | 0.126 |
| Yes | 21 (1.1) | 9 (1.6) | 2 (0.4) | 7 (1.2) | 3 (1.3) | |
| No | 2253 (98.9) | 647 (98.4) | 597 (99.6) | 488 (98.8) | 521 (98.7) | |
| **Healthy Eating Index (mean (SD))** | 62.52 (12.66) | 59.61 (11.78) | 64.14 (12.47) | 64.25 (13.14) | 62.11 (12.66) | <0.001 |

*(Continued)*

**Table 1.** (Continued)

| | Selenium intake (μg/day) | | | | | P-value |
|---|---|---|---|---|---|---|
| | Overall | <78.0 | 78.0–111.5 | 111.6–150.9 | >150.9 | |
| **Serum selenium (mean (SD))** | 125.59 (17.39) | 124.49 (20.96) | 124.28 (15.52) | 126.81 (17.38) | 126.69 (14.96) | 0.289 |

Abbreviations: NAFLD, nonalcoholic fatty liver disease; NHANES III, the Third National Health and Nutrition Examination Survey.

All estimates accounted for complex survey designs, and all percentages were weighted. Data are presented as mean ± SD or n (%).

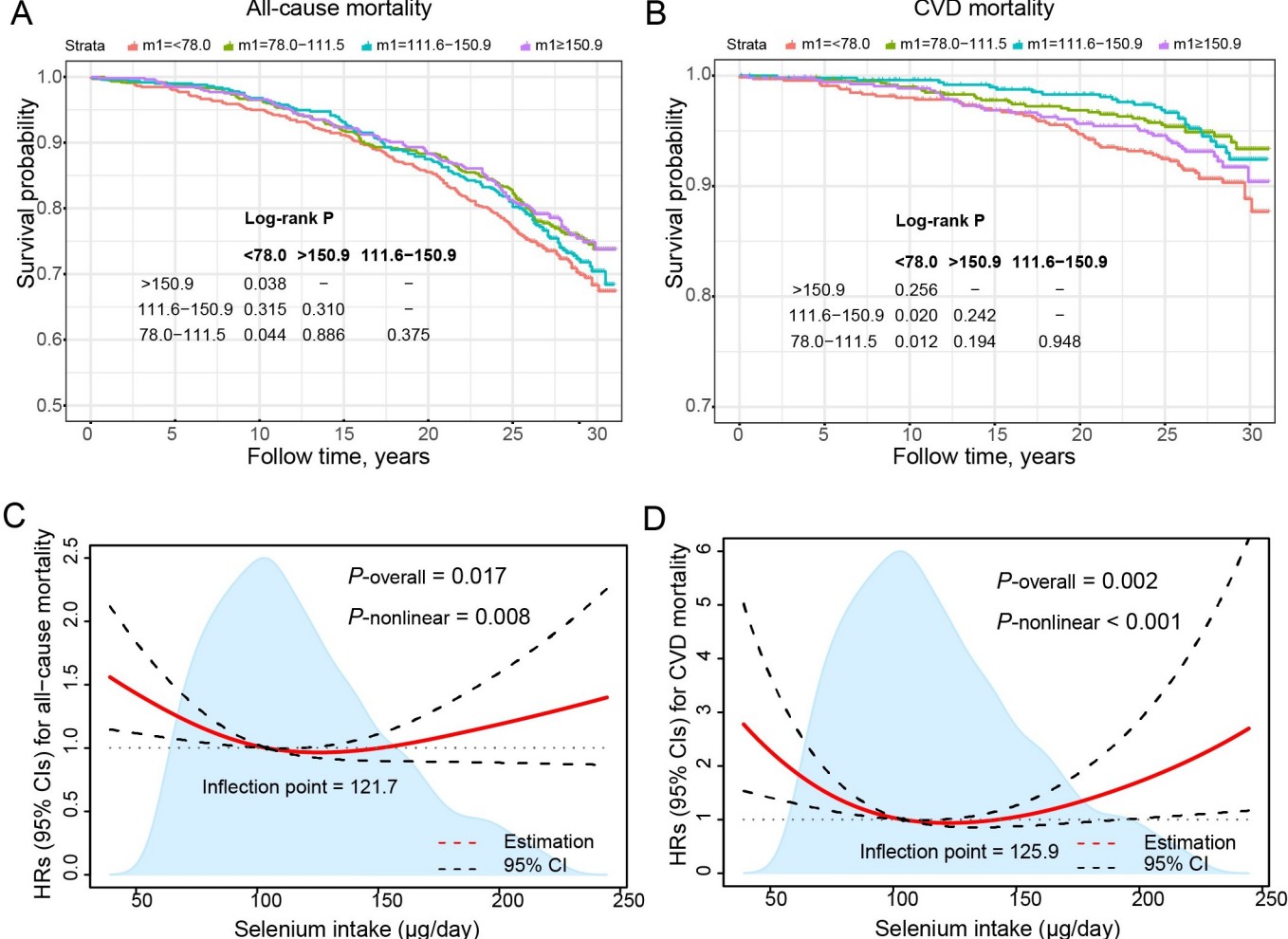

**Fig 2.** Kaplan-Meier survival curves of selenium intake with all-cause (A) and cardiovascular (B) mortality. Association between selenium intake and all-cause (C) and cardiovascular mortality (D) among patients with NAFLD. Cox proportional hazards models were used to estimate the HRs (95% CIs) for mortality according to selenium intake. Adjusted for age (years), sex (male or female), and self-reported race (non-Hispanic White, non-Hispanic Black, Mexican American, or others), education (less than high school, high school or equivalent, or college or above), Healthy Eating Index (continuous), family income-poverty ratio (≤1.30, 1.31–3.50, or >3.50), physical activity (inactive, insufficiently active, or active), smoking status (never, former, or current smoker), body mass index (kg/m$^2$; <25.0, 25.0–29.9, or ≥30.0), diabetes (yes or no), hypertension (yes or no), dyslipidemia (yes or no), Fibrosis-4 index (<1.30, 1.30–2.66, or ≥2.67). NAFLD, nonalcoholic fatty liver disease; NHANES III, the Third National Health and Nutrition Examination Survey; CIs, confidence intervals; HRs, hazard ratios.

**Table 2. Hazard ratios (95% CIs) for mortality according to selenium intake (quartiles) for participants with NAFLD in NHANES III.**

| | Selenium intake (μg/day) | | | | $P_{\text{trend}}$ | One-unit increment in log-transformed selenium intake |
|---|---|---|---|---|---|---|
| | <78.0 | 78.0–111.5 | 111.6–150.9 | >150.9 | | |
| **All-cause mortality** | | | | | | |
| No. deaths/total | 188/656 | 140/599 | 129/495 | 120/524 | | |
| **Model 1** | 1 | 0.71 (0.52–0.98) | 0.7 (0.48–1.02) | 0.66 (0.49–0.9) | 0.021 | 0.69 (0.45–1.06) |
| HR (95% CI) | | | | | | |
| *P*-value | | 0.035 | 0.067 | 0.008 | | 0.087 |
| **Model 2** | 1 | 0.61 (0.43–0.87) | 0.63 (0.39–1.01) | 0.67 (0.48–0.95) | 0.088 | 0.70 (0.44–1.12) |
| HR (95% CI) | | | | | | |
| *P*-value | | 0.006 | 0.053 | 0.025 | | 0.135 |
| **Cardiovascular mortality** | | | | | | |
| No. deaths | 58 | 31 | 26 | 37 | | |
| **Model 1** | 1 | 0.26 (0.14–0.49) | 0.50 (0.22–1.14) | 0.47 (0.21–1.04) | 0.189 | 0.5 (0.18–1.34) |
| HR (95% CI) | | | | | | |
| *P*-value | | <0.001 | 0.099 | 0.063 | | 0.166 |
| **Model 2** | 1 | 0.19 (0.08–0.45) | 0.29 (0.12–0.68) | 0.38 (0.17–0.85) | 0.126 | 0.28 (0.09–0.91) |
| HR (95% CI) | | | | | | |
| *P*-value | | <0.001 | 0.005 | 0.018 | | 0.035 |

Abbreviations: NAFLD, nonalcoholic fatty liver disease; NHANES III, the Third National Health and Nutrition Examination Survey; CIs, confidence intervals; HRs, hazard ratios.

Cox proportional hazards models were used to estimate the HRs (95% CIs) for mortality according to selenium intake (quartiles).

Model 1 Adjusted for age (years), sex (male or female), and self-reported race (non-Hispanic White, non-Hispanic Black, Mexican American, or others).

Model 2 Further adjusted for education (less than high school, high school or equivalent, or college or above), Healthy Eating Index (continuous), family income-poverty ratio (≤1.30, 1.31–3.50, or >3.50), smoking status (never, former, or current smoker), physical activity (inactive, insufficiently active, or active), body mass index (kg/m$^2$; <25.0, 25.0–29.9, or ≥30.0), dyslipidemia (yes or no), diabetes (yes or no), hypertension (yes or no), Fibrosis-4 index (<1.30, 1.30–2.66, or ≥2.67).

insignificant. Of note, no significant interactions ($P_{\text{interaction}} > 0.05/8$) were found between selenium intake and stratified variables except for the variable of any type of comorbidities (yes or no).

## Sensitivity analyses

In the sensitivity analyses, similar results were observed when excluding individuals with a baseline history of heart failure or stroke (S2 Table), and when excluding individuals who died within the first 5 years of follow-up (S3 Table).

## Discussion

In this prospective study of nationally representative United States adults, we found U-shaped associations between selenium intake with all-cause and cardiovascular mortality among adults with NAFLD, with inflection points of 121.7 μg/day and 125.9 μg/day, respectively. Moreover, participants with higher dietary selenium intake were positively associated with higher levels of LDL-C and HOMA-IR.

Selenium is an essential nutrient for human health, but its effects on human health are multifaceted and complex [21]. Selenium is incorporated into selenium proteins, such as selenium-dependent glutathione peroxidases, has a wide range of pleiotropic properties, and plays an important role in oxidative stress and inflammatory immune responses [21]. A number of

**Table 3. Threshold effect analysis of selenium intake on all-cause and cardiovascular mortality among participants with NAFLD using the 2-piecewise Cox proportional hazards model in NHANES III.**

|  | HR (95% CI) | *P*-value |
|---|---|---|
| **All-cause mortality** |  |  |
| Fitting by the standard Cox regression model | 1.00 (0.99–1.00) | 0.287 |
| Fitting by two-piecewise Cox regression model |  |  |
| **Inflection point** | 121.7 μg/day |  |
| Selenium intake ≤ 121.7 μg/day | 0.99 (0.99–1.00) | 0.009 |
| Selenium intake > 121.7 μg/day | 1.00 (1.00–1.01) | 0.143 |
| P for Log-likelihood ratio | 0.018 |  |
| **Cardiovascular mortality** |  |  |
| Fitting by the standard Cox regression model | 1.0 (0.99–1.00) | 0.552 |
| Fitting by two-piecewise Cox regression model |  |  |
| **Inflection point** | 125.9 μg/day |  |
| Selenium intake ≤ 125.9 μg/day | 0.98 (0.98–0.99) | 0.001 |
| Selenium intake > 125.9 μg/day | 1.01 (1.00–1.02) | 0.007 |
| P for Log-likelihood ratio | <0.001 |  |

Abbreviations: NAFLD, nonalcoholic fatty liver disease; NHANES III, the Third National Health and Nutrition Examination Survey; CIs, confidence intervals; HRs, hazard ratios.

Two-piecewise Cox proportional hazards model were used to threshold effect analysis of selenium intake on all-cause and cardiovascular mortality among participants with NAFLD. Adjusted for age (years), sex (male or female), and self-reported race (non-Hispanic White, non-Hispanic Black, Mexican American, or others), education (less than high school, high school or equivalent, or college or above), Healthy Eating Index (continuous), family income-poverty ratio (≤1.30, 1.31–3.50, or >3.50), smoking status (never, former, or current smoker), physical activity (inactive, insufficiently active, or active), body mass index (kg/m$^2$; <25.0, 25.0–29.9, or ≥30.0), dyslipidemia (yes or no), diabetes (yes or no), hypertension (yes or no), Fibrosis-4 index (<1.30, 1.30–2.66, or ≥2.67).

animal experiments have shown that dietary selenium supplementation can effectively prevent liver damage and insulin resistance during the development of NAFLD in mice fed a high-fat diet [22, 23]. This may be related to the reduction of selenium levels during the progression of NAFLD, and selenium supplementation can effectively alleviate metabolic disorders by alleviating antioxidant stress and anti-inflammatory regulation [24].

Despite the nutritional value of selenium supplements, there is growing concern about selenium supplements, especially in light of the termination of the Selenium and Vitamin E cancer prevention trial, where high rates of diabetes and high-grade prostate cancer were found in the selenium supplement group [25], which may underscore the fact that supplements only produce benefits if nutrient intake is insufficient, and that additional selenium supplementation in people who already have enough selenium may increase disease risk. Several epidemiological studies have evaluated the association between selenium and NAFLD [26–28]. In a large cross-sectional study of 8,550 Chinese adults, a positive log-linear dose-response relationship between serum selenium and the prevalence of NAFLD was observed [26]. In a cross-sectional study of 3,827 people in United States, serum selenium levels above 130ug/L were found to be positively associated with the prevalence of NAFLD [27]. Another cross-sectional study of 5,436 middle-aged and elderly Chinese also reported a positive association between dietary selenium intake and the prevalence of NAFLD [28]. Our findings showed U-shaped associations between dietary selenium intake and all-cause and cardiovascular mortality in patients with NAFLD, suggesting that both high and low selenium intake could be detrimental to patients with NAFLD.

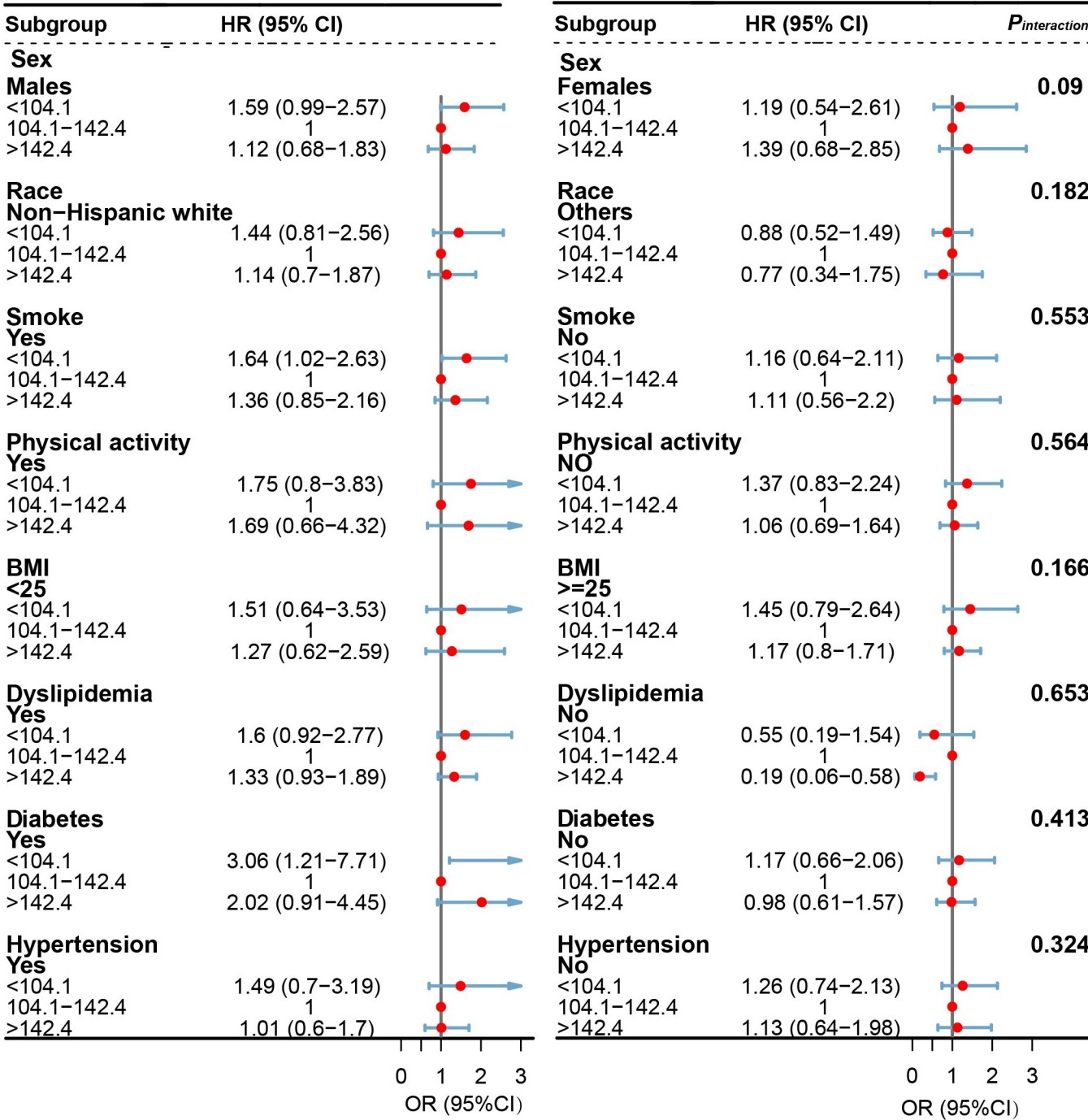

**Fig 3. Forest plots of stratified analyses of selenium intake and all-cause mortality among patients with NAFLD.** Cox proportional hazards models were used to estimate the HRs (95% CIs) for mortality according to selenium intake (<104.1 vs. 104.1–142.4 vs. >142.4 μg/day). Adjusted for age (years), sex (male or female), and self-reported race (non-Hispanic White, non-Hispanic Black, Mexican American, or others).education (less than high school, high school or equivalent, or college or above), Healthy Eating Index (continuous), family income-poverty ratio (≤1.30, 1.31–3.50, or >3.50), physical activity (inactive, insufficiently active, or active), smoking status (never, former, or current smoker), body mass index (kg/m$^2$; <25.0, 25.0–29.9, or >30.0), dyslipidemia (yes or no), diabetes (yes or no), hypertension (yes or no), Fibrosis-4 index (<1.30, 1.30–2.66, or ≥2.67). NAFLD, nonalcoholic fatty liver disease; NHANES III, the Third National Health and Nutrition Examination Survey; CIs, confidence intervals; HRs, hazard ratios.

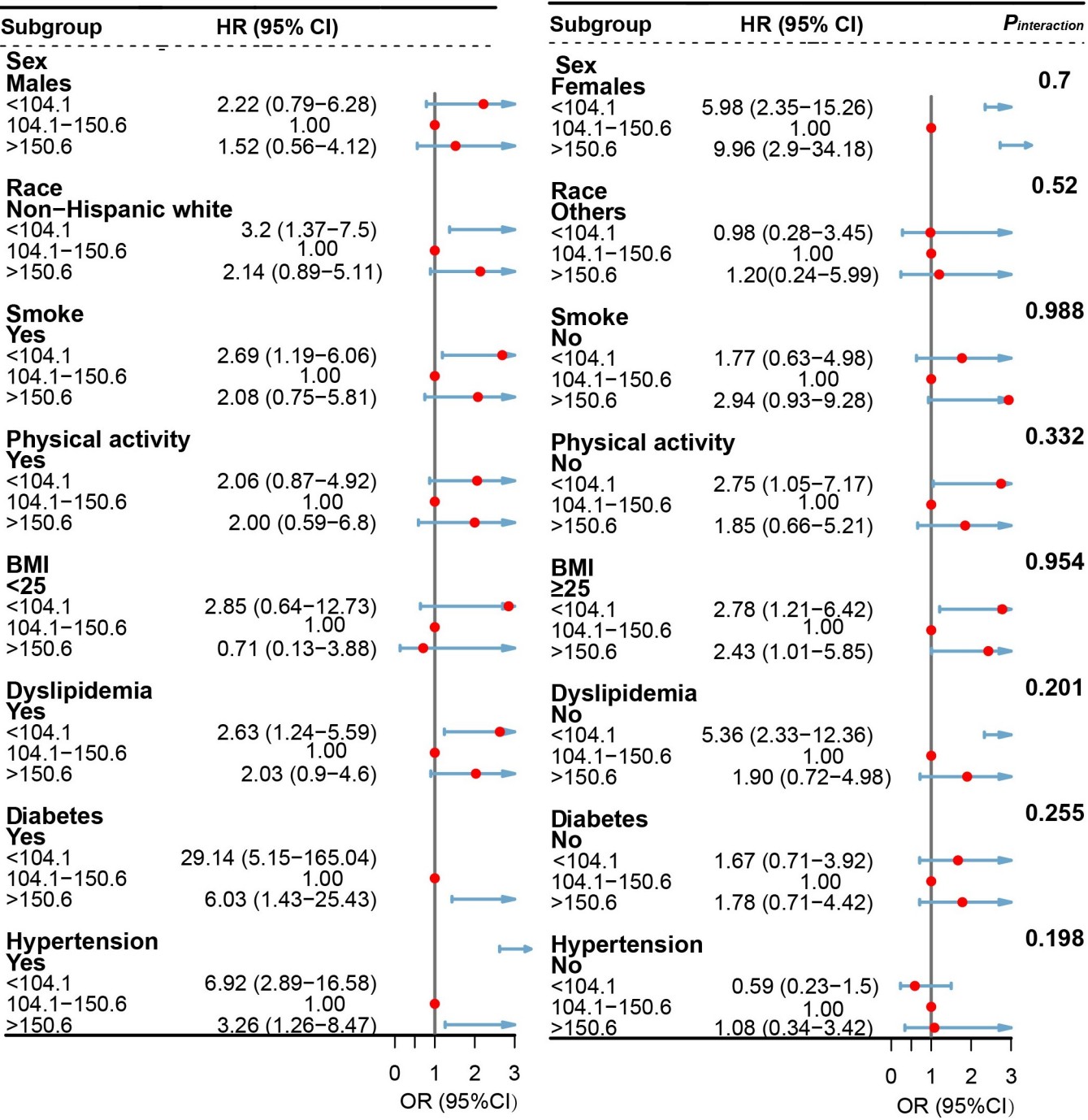

**Fig 4. Forest plots of stratified analyses of selenium intake and cardiovascular mortality among patients with NAFLD.** Cox proportional hazards models were used to estimate the HRs (95% CIs) for mortality according to selenium intake (<104.1 vs. 104.1–121.7 vs. >121.7 μg/day). Adjusted for age (years), sex (male or female), and self-reported race (non-Hispanic White, non-Hispanic Black, Mexican American, or others).education (less than high school, high school or equivalent, or college or above), Healthy Eating Index (continuous), family income-poverty ratio (≤1.30, 1.31–3.50, or >3.50), physical activity (inactive, insufficiently active, or active), smoking status (never, former, or current smoker), body mass index (kg/m$^2$; <25.0, 25.0–29.9, or ≥30.0), dyslipidemia (yes or no), diabetes (yes or no), hypertension (yes or no), Fibrosis-4 index (<1.30, 1.30–2.66, or ≥2.67). NAFLD, nonalcoholic fatty liver disease; NHANES III, the Third National Health and Nutrition Examination Survey; CIs, confidence intervals; HRs, hazard ratios.

The potential mechanism by which dietary selenium intake affects NAFLD progression is still unclear and deserves to be explored. The relationship between dietary selenium intake and NAFLD may be mediated by inducing insulin resistance and oxidative stress, which play an important role in the occurrence and development of NAFLD [29]. For example, previous in vivo animal studies have shown that diets high in selenium or selenium supplements are associated with increased expression of glutathione peroxisase-1 or other antioxidant selenium proteins that can induce insulin resistance in the liver [30, 31]. In addition, selenium compounds, especially selenite, lead to the production of toxic reactive oxygen species, and the enhanced production of reactive oxygen species can further cause lipid peroxidation by activating stellate cells, which can lead to liver inflammation and cardiovascular disease [32]. These findings were consistent with our results that selenium intake was positively associated with levels of LDL-C and HOMA-IR.

In addition to having many cardiometabolic effects, selenium proteins prevent oxidative modification of lipids, inhibit platelet aggregation, and reduce inflammation, evidence supporting the potential cardiovascular benefits of selenium. However, randomized trials of selenium-containing supplements have not shown a significant protective effect against cardiovascular disease or mortality endpoints [33–35]. Combining previous studies with our findings, it is suggested that appropriate selenium supplementation can be recommended for NAFLD patients with insufficient selenium to reduce the progression of the disease, while dietary selenium supplementation is not recommended for patients with sufficient selenium.

One of the main strengths of the current study is the use of a nationally representative sample of United States NAFLD patients with a long follow-up period, which helps to generalize our results to the population. Second, cause-of-death information is obtained by linking to national death index records. In addition, the comprehensive data from NHANES allowed us to adjust for a wide range of confounding factors, such as lifestyle factors, socioeconomic status, race/ethnicity, and comorbidities.

Several limitations of our study should be noted. First, although 24-hour recall techniques are widely used in epidemiological studies and surveys, it is clear that such recall is subject to bias, and often they underestimate actual dietary intake. Second, due to the study design, we were unable to directly infer a causal relationship between selenium intake and mortality in patients with NAFLD. Finally, although we controlled for many possible confounders, our findings may have been influenced by residual confounders or random errors.

## Conclusions

This prospective cohort study illustrated U-shaped associations between selenium intake and all-cause and cardiovascular mortality among adults with NAFLD, suggesting that both too high and too low selenium intake increase the risk of death among patients with NAFLD. Our findings help provide dietary guidance to patients with NAFLD to mitigate disease progression and improve prognosis.

## Supporting information

**S1 Table. Least squares means of cardiac and liver metabolic markers according to selenium intake among participants with NAFLD.**
(DOCX)

**S2 Table. Least squares means of cardiac and liver metabolic markers according to selenium intake among participants with NAFLD.**
(DOCX)

**S3 Table. HRs (95% CIs) of mortality according to selenium intake after excluding participants with NAFLD who died within the first five years of follow-up in NHANES III.**
(DOCX)

## Acknowledgments

We thank Centers for Disease Control and Prevention (CDC) in the United States for organizing this study. And we thank for all generous voluntary participation of the United States residents who have given their personal time to make the NHANES surveys possible.

## Author Contributions

**Conceptualization:** Xin Dong.

**Data curation:** Xin Dong, Yunchao Deng.

**Formal analysis:** Xin Dong.

**Investigation:** Xin Dong.

**Methodology:** Xin Dong, Gang Chen.

**Project administration:** Gang Chen.

**Software:** Xin Dong, Yunchao Deng.

**Supervision:** Gang Chen.

**Validation:** Xin Dong.

**Writing – original draft:** Xin Dong.

**Writing – review & editing:** Gang Chen.

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
