## [Decision Letter · Decision Letter 0]

27 Feb 2024

PONE-D-23-33614Selenium intake in relation to all-cause and cardiovascular mortality in individuals with nonalcoholic fatty liver disease:a nationwide  study in NutritionPLOS ONE

Dear Dr. Chen,

Thank you for submitting your manuscript to PLOS ONE. After careful consideration, we feel that it has merit but does not fully meet PLOS ONE’s publication criteria as it currently stands. Therefore, we invite you to submit a revised version of the manuscript that addresses the points raised during the review process.

We look forward to receiving your revised manuscript.

Kind regards,

Aneta Agnieszka Koronowicz, PhD

Academic Editor

PLOS ONE

Journal Requirements:

**Additional Editor Comments:**

Major revisions are necessary to improve the paper.  You have to revise the paper in accordance with reviewers comments. Please create an "author response" file with a point-by-point response to each comment, clearly describing how they have been addressed in the revision.

Reviewers' comments:

Reviewer's Responses to Questions

**Comments to the Author**

1. Is the manuscript technically sound, and do the data support the conclusions?

Reviewer #1: Yes

Reviewer #2: Yes

2. Has the statistical analysis been performed appropriately and rigorously? 

Reviewer #1: Yes

Reviewer #2: Yes

3. Have the authors made all data underlying the findings in their manuscript fully available?

Reviewer #1: Yes

Reviewer #2: Yes

4. Is the manuscript presented in an intelligible fashion and written in standard English?

Reviewer #1: Yes

Reviewer #2: Yes

5. Review Comments to the Author

Reviewer #1: Thanks a lot to respected authors for this valuable study.

The study was well designed and had appropriate statistical analysis.

You can insert the amount of p-value on the kaplan-meier plot for better visualization.

Reviewer #2: I would like to express my gratitude to the author for conducting this valuable research. However, I have some comments:

1. Please clearly mention the previous literature regarding this subject in the introduction and provide a rationale for the study. The novelty of the work should be highlighted, particularly in comparison with a previously published study entitled "Increased serum selenium levels are associated with reduced risk of advanced liver fibrosis and all-cause mortality in NAFLD patients: National Health and Nutrition Examination Survey (NHANES) III" (doi: 10.1016/j.aohep.2020.07.006).

2. Please clearly state the primary and secondary objectives in the last part of the introduction. While it appears that the study has focused on mortality, there are also other outcomes discussed in the manuscript (such as LDL, HOMA, etc.).

3. Please conduct a posttest power analysis to ensure the validity of your results.

6. PLOS authors have the option to publish the peer review history of their article (what does this mean?). If published, this will include your full peer review and any attached files.

Reviewer #1: No

Reviewer #2: No

---

## [Author Response · Author response to Decision Letter 0]

30 Mar 2024

Dear Editor,

We would like to thank the reviewers for their recognition and suggestions on our research and paper, and the editor for giving us the opportunity to revise the manuscript. We carefully revised the manuscript according to the comments of the reviewers, answered the questions of each reviewer, and upload my revised manuscript together with the version with the tracking modification mark as a supplementary file.

Reviewer #1

Thanks to Reviewer #1 for patiently reviewing my manuscript and providing constructive comments. We have inserted the amount of p-value on the Kaplan-Meier plot for better visualization.

Reviewer #2

We would like to thank Reviewer #2 for the recognition and suggestions on this article, which helped us to supplement the background knowledge and improve the quality of the manuscript.

1. We have quoted Reja et al. 's study “Increased serum selenium levels are associated with reduced risk of advanced liver fibrosis and all-cause mortality in NAFLD patients: National Health and Nutrition Examination Survey (NHANES) III” in the introduction of the article and indicated the novelty and significance of our study. This study mainly reflected the effect of blood selenium homeostasis on liver fibrosis in patients with NAFLD, rather than dietary selenium intake. However, the associations and dose-response relationships between dietary selenium intake and all-cause and cardiovascular mortality in individuals with NAFLD are unclear. Our prospective cohort study illustrated U-shaped associations between selenium intake and all-cause and cardiovascular mortality among adults with NAFLD, suggesting that both too high and too low selenium intake increase the risk of death among patients with NAFLD. Our findings help provide dietary guidance to patients with NAFLD to mitigate disease progression and improve prognosis.

2. We have added a description of the primary and secondary objectives in the last part of the introduction. The primary objective of this study was to investigate the association between dietary selenium intake with all-cause and cardiovascular mortality in adults under than 60 years of age with NAFLD using the NHANES III collected from 1988-1994. The secondary goal is to determine the optimal range for dietary selenium intake, which may help guide dietary adjustments in patients with NAFLD. The third objective is to investigate the effects of dietary selenium intake on glucose metabolism and lipid metabolism in adults under 60 years of age with NAFLD.

3. We have added power analysis to least squares means of cardiac and liver metabolic markers according to selenium intake among participants with NAFLD to ensure the validity of our results.

Finally, once again, we would like to thank the reviewers for their recognition and suggestions on our research and paper, and the editor for the opportunity to revise our manuscript. We refined the manuscript with suggestions from the reviewers and uploaded the latest manuscript in the system. We look forward to your reply. If you have any questions, please feel free to contact me.

Thanks.

Sincerely,

Corresponding author：Gang Chen, M.M.

Institution：Department of Cardiovascular Medicine, The Central Hospital of Wuhan, Tongji Medical College, Huazhong University of Science and Technology, Wuhan 430014, China.

Email: tomgangchen@sohu.com

---

## [Decision Letter · Decision Letter 1]

23 Apr 2024

Selenium intake in relation to all-cause and cardiovascular mortality in individuals with nonalcoholic fatty liver disease:a nationwide  study in nutrition

PONE-D-23-33614R1

Dear Dr. Chen,

We’re pleased to inform you that your manuscript has been judged scientifically suitable for publication and will be formally accepted for publication once it meets all outstanding technical requirements.

Kind regards,

Aneta Agnieszka Koronowicz, PhD

Academic Editor

PLOS ONE

Additional Editor Comments (optional):

Reviewers' comments:

Reviewer's Responses to Questions

**Comments to the Author**

1. If the authors have adequately addressed your comments raised in a previous round of review and you feel that this manuscript is now acceptable for publication, you may indicate that here to bypass the “Comments to the Author” section, enter your conflict of interest statement in the “Confidential to Editor” section, and submit your "Accept" recommendation.

Reviewer #2: (No Response)

2. Is the manuscript technically sound, and do the data support the conclusions?

Reviewer #2: (No Response)

3. Has the statistical analysis been performed appropriately and rigorously? 

Reviewer #2: (No Response)

4. Have the authors made all data underlying the findings in their manuscript fully available?

Reviewer #2: (No Response)

5. Is the manuscript presented in an intelligible fashion and written in standard English?

Reviewer #2: (No Response)

6. Review Comments to the Author

Reviewer #2: (No Response)

7. PLOS authors have the option to publish the peer review history of their article (what does this mean?). If published, this will include your full peer review and any attached files.

Reviewer #2: No

---

## [Editor Report · Acceptance letter]

30 Apr 2024

PONE-D-23-33614R1 

PLOS ONE

Dear Dr. Chen, 

I'm pleased to inform you that your manuscript has been deemed suitable for publication in PLOS ONE. Congratulations! Your manuscript is now being handed over to our production team.

Kind regards, 

on behalf of

Prof. Aneta Agnieszka Koronowicz 

Academic Editor

PLOS ONE